

# GC Insights: Communicating changes in local climate risk using a physically plausible causal chain

**Ed Hawkins[1], Nigel Arnell[2], Jamie Hannaford[3] and Rowan Sutton[1]**

[1] National Centre for Atmospheric Science, Department of Meteorology, University of Reading, Reading, UK

[2] Department of Meteorology, University of Reading, Reading, UK

[3] Centre for Ecology and Hydrology, Wallingford, UK

*Correspondence to*: Ed Hawkins (ed.hawkins@ncas.ac.uk)

**Abstract.** Directly linking greenhouse gas emissions or global warming to experiences of local climatic changes or extreme

events is a potentially important communication tool. We develop a physically plausible 'causal chain' as one approach to demonstrate the connections between global carbon dioxide emissions and real-life events, using a case study of flood risk in one river basin in the UK.

It is unequivocal that human activities are warming the climate (IPCC, 2021). This type of global assessment is critical for those needing to make decisions on mitigating against the risks of ongoing climatic changes but is potentially less relevant when communicating to individuals about how climate change matters to them.

The effects of climate change will often feel most 'real' during an extreme event, such as a flood or heatwave, especially when

significant harm is caused. The process of linking an individual extreme weather event to climate change – event attribution – is now a commonly used and effective tool, often communicated to decision-makers, media and the public in near-real-time (e.g. van Oldenborgh et al., 2021). There is some evidence that attribution of weather events that are experienced personally can generate climate change concern and changed decisions, but this effect may be limited, especially if the attribution conflicts with an individual's prior beliefs (Sambrook et al. 2021).


Here we focus on communicating long-term trends in local climate risks. In this case, it might be people's experience that certain types of extreme event are becoming more frequent, or that the consequences are getting worse, for example. We develop a 'causal chain' as a communication approach to highlight how global carbon dioxide emissions, which may feel rather abstract to any individual, affect climate risks that matter to people. To ensure relevance and improve understanding we

consider it is essential for the causal chain to be based on observations and well understood physical principles rather than, for example, the output of complex climate models.



We consider flood risk in one UK river basin as the end-point of the causal chain, but this approach could be developed more generally for other types of climate risk and in other countries. The UK is fortunate to have lengthy observation-based datasets of several relevant climate variables available for open use (Hollis et al. 2019).


Figure 1 shows simple observation-based timeseries of climate-related changes using the full extent of the available records. Cumulative anthropogenic global carbon dioxide emissions since 1750 have exceeded 2500 GtCO2 (grey), around 70% of which are due to burning fossil fuels (Friedlingstein et al. 2023). These emissions, along with those of other greenhouse gases and anthropogenic factors, have caused global temperatures to increase by around 1.2°C since the era before widespread

industrialisation (orange; Morice et al. 2021). The best estimate is that human activities have caused all the observed warming, and that global warming will stop when we reach net-zero anthropogenic carbon dioxide emissions, i.e. when cumulative emissions stop increasing (IPCC, 2021).

Global warming is experienced in the UK as a slightly larger change in mean temperature (1.4°C since 1884; red); land areas

warm faster than ocean areas (Byrne & O'Gorman, 2018). There is a corresponding increase in near-surface specific humidity (+7.7% since 1960; green) as a warmer atmosphere can hold more moisture due to the Clausius-Clapeyron relationship. In turn, this increase in atmospheric water content leads to more intense rainfall (+14% since 1891; blue) – when it rains, it rains more (e.g. King et al. 2023). This observed increase in UK rainfall intensity occurs in all seasons but is larger in winter than in summer (not shown).


The dashed black lines show the regression of a smoothed version of the global temperature series onto the other observed timeseries. Note particularly that the signature of the ups and downs of global temperature are also visible in the local, noisier timeseries over the UK. For example, there is a slight cooling of UK temperatures, and small reductions in rainfall intensity, during the 1950s-1960s when global temperatures also slightly cool. This suggests that variations in global temperature are

being directly experienced in the UK; it is notable that the rainfall intensity data is entirely independent from the global temperature data.

Although rainfall intensity could be the end-point in this causal chain, we take a further step to relate the causes of global warming to local impacts and risks by considering the effects on a single river basin. In this example, the annual maximum

peak flow for the Ribble river has increased by 44% over the past 50 years (black; NRFA, 2023). If you live near the Ribble, this is of direct relevance to your own personal flood risk; this river basin experienced severe floods affecting hundreds of homes in the 1980-1981, 2015-2016 and 2019-2020 'water years' (October to September) when the peak flows were largest.



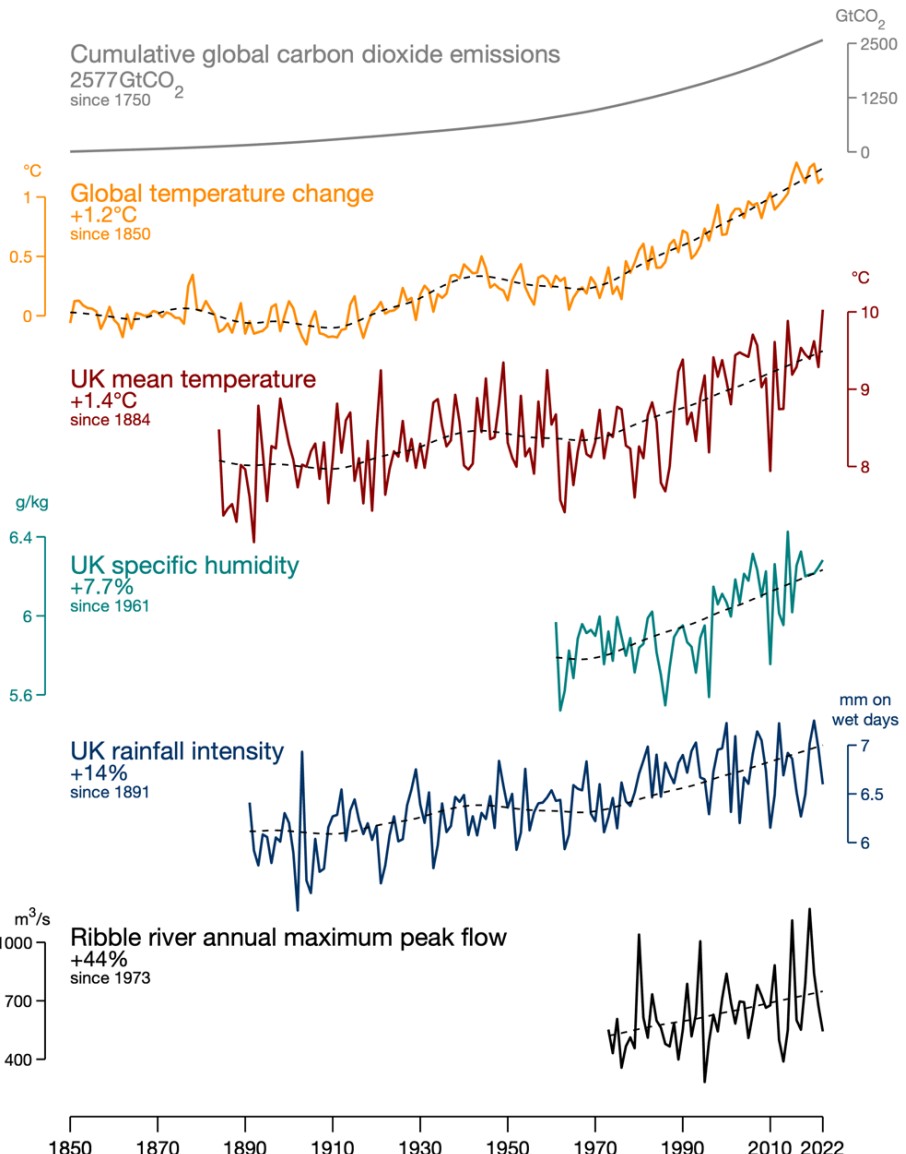

**Figure 1: An observation-based causal chain from global carbon dioxide emissions to peak flow in the Ribble river.**
Cumulative anthropogenic global carbon dioxide emissions (grey; Friedlingstein et al. 2023) are causing global temperatures to increase (orange, with 41-year loess smoothed version in dashed black; Morice et al. 2021). Global warming is experienced locally as warmer UK temperatures (red), increased specific humidity (green), with corresponding increases in rainfall intensity (blue); all based on data from Hollis et al. (2019). Annual maximum peak flow on the Ribble river has increased in the last 50 years (black; NRFA 2023). Black dashed lines show regressions onto the smoothed global temperature timeseries.



One potential implication of this causal chain is that global emissions of carbon dioxide directly increase the risk of flooding from the Ribble river, but there is no formal attribution for that conclusion. However, we note that the links in this causal chain are well studied. The increase in global temperature due to cumulative carbon dioxide emissions has been quantified (IPCC, 2021), and UK temperature rise has been attributed to anthropogenic factors and global warming (Karoly & Stott, 2006;

Hawkins et al. 2020). Increases in specific humidity have been attributed to human influences globally (Willett et al. 2007), and the physical reasons why warming will cause increases in humidity and rainfall intensity are well established (Pfahl et al. 2017), including the Clausius-Clapeyron relationship. A human influence on the risk of specific UK flood events has also been demonstrated (Pall et al. 2011; Schaller et al. 2016; Otto et al. 2018).

But, what about the final link in the chain? There is a robust increase in peak river flows in many UK regions (Hannaford et al. 2021, Slater et al. 2021) and, all else being equal, more intense rainfall will lead to more runoff, higher peak flows and increased flood risk. However, for any particular river basin, this final link in the chain is highly complex due to the role of catchments in modulating changes in rainfall variability; all else is rarely equal (Hannaford, 2015). The complexities can include the role of antecedent conditions and evaporation, catchment storage in groundwater or soils that could dampen

extreme rainfall increases, and direct anthropogenic modifications, such as reservoirs or river engineering. As such, there is often a highly non-linear relationship between trends in extreme rainfall and river flooding, and trends in the former do not always lead to similar responses in the latter (Do et al., 2020).

In our chosen example, the Ribble is a 'benchmark' catchment (Harrigan et al. 2018) as it is largely free of major disturbances

and is relatively responsive to rainfall variability given the wetness of the setting, upland terrain and impermeable geology. Hence, the above complexities are likely to be minimised, and there is reasonable confidence in the link between extreme rainfall and flow responses, although there is a possible role for other local factors such as land use change.

These relatively simple connections between global carbon dioxide emissions, increases in global average temperature and

severe impacts on people could be a useful tool to develop further. We suggest that talking through the links in this causal chain may be useful for local decision makers and the millions of people living with increased flood risk in the UK, such as those in the Ribble valley, to understand how climate change may directly affect them and inform decisions on adaptation.

Long observation-based records are extremely useful for communicating that the climate has changed and how this is already

affecting people. The extension of this approach to other climate-related hazards such as heatwaves, droughts, storm surges, or wildfires may also be useful, along with expanding to other locations or countries, dependent on data availability. Further work could also develop more complex causal networks or storylines for some of the links in the chain (Niemeijer & de Groot, 2008; Shepherd et al. 2018). However, as with river flows, it will undoubtedly be the last link, towards the impact, that will bring the most complexity.



## Competing interests

Ed Hawkins is an Editor for Geoscience Communication.

### Acknowledgements

EH and RS are supported by the UK National Centre for Atmospheric Science. This publication has emanated from research conducted with the financial support from a Co-Centre award number 22/CC/11103. The Co-Centre award is managed by Science Foundation Ireland (SFI), Northern Ireland's Department of Agriculture, Environment and Rural Affairs (DAERA) and UK Research and Innovation (UKRI), and is supported via UK's International Science Partnerships Fund (ISPF), and the Irish Government's Shared Island initiative.

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
