# Peer review of "GC Insights: Communicating long-term changes in local climate risk using a physically plausible causal chain"

_EGUsphere, 2024_

## Author Response (AR2)

**GC Insights: Communicating changes in local climate risk using a physically plausible causal chain**

**Response to Executive Editor**

We have made some small changes to the text as technical corrections. The word 'visualisation' has been added in the abstract, and edits have been made to the paragraph starting on L25 to highlight the novel aspects. The have added a short sentence on L103 to mention that we will examine the efficacy of the graphic on future.

**Response to Editor**

*Regarding RC1 "Line 49: even though the "[…] observed increase in UK rainfall intensity […] is larger in winter than in summer" is not shown in Figure1, a reference is still needed in the text", in your response, you mention that this has not been shown for winter and summer separately before. I agree with the reviewer that a reference would be beneficial for such statement, and I wonder if you could rephrase the sentence to say that UK winters are getting wetter (without the mention of summers), a statement which you can back up with a reference.*

We agree a citation would be helpful but the specific statement is about rainfall intensity (mm on wet days). As far as we are aware, only an annual average rainfall intensity timeseries has been published before (including in the Met Office State of the Climate annual reports) so we can't cite any seasonal references. Although winter mean rainfall has increased, this is not the same metric.

*I suggest adding a data availability section at the end of your manuscript that links to the GitHub repository you created.*

Done. Thank you.

Additionally, you could consider sharing the code you developed for plotting Fig. 1 alongside the data.

After careful consideration, we have chosen not to make the code available openly as it is just plotting timeseries and written specifically for this application. We would be happy to supply the MATLAB code if anyone really wants it!

**Edits**

We have added the following sentence to the start of the fourth paragraph to highlight the flood impacts a little more:
*Widespread recent flood impacts have led to increasing concerns over changing UK flood regimes in a warming world (e.g. Hannaford et al. 2021 and references therein).*

We have also added a reference to Kew et al. (2024), a recent extensive report on the impact of climate change on recent UK floods.

**Response to reviewers**

**Reviewer 1:**

*The paper presents an approach to communicate variations in local climate risks using a physically causal chain providing the example of a river in the UK. Overall, the manuscript follows the characteristics of a GC Insights paper (i.e., the title starts with "GC Insights", the number of words falls within the range indicated for GC Insights, the abstract is brief, there is one figure and no tables). The paper is well-written, the title reflects the content of the paper and references seem appropriate. I appreciate the fact that authors relied on observations of well-known physical entities. A competing interest is clearly stated.*

We thank the reviewer for their careful reading of the manuscript and constructive comments.

*Detailed comments on the current version of the manuscript:*

*Title: I would suggest adding "…Communicating long-term changes…" to better address the content of the manuscript.*

We agree with the reviewer and will add 'long-term' to the title.

*The GC Insights format indicates that "[…] any conclusions should also be evident from the Title and Abstract", thus I would suggest adding to the abstract a sentence clearly dedicated to the conclusions of the work.*

We agree and have revised the abstract slightly:
*Directly linking greenhouse gas emissions or global warming to experiences of local climatic changes is a potentially important communication tool. Using observations we develop a physically plausible 'causal chain' to demonstrate the connections between global carbon dioxide emissions and local climate events. We highlight how increased flood risk in one river basin in the UK could be discussed with people directly affected by recent floods.*

*Line 16: "potentially less relevant" might be misleading. The link between human activities and warming climate might be difficult to communicate to certain audiences, but still (could) be relevant also to individuals. So, I would suggest rewriting this sentence.*

We think the use of 'potentially' is a strong enough caveat, but the sentence has been tweaked:
*This type of global assessment is critical for those needing to make decisions on mitigating against the risks of ongoing climatic changes but is potentially less relevant when communicating to individuals about how climate change may directly affect them.*

*Line 49: even though the "[...] observed increase in UK rainfall intensity [...] is larger in winter than in summer" is not shown in Figure1, a reference is still needed in the text.*

The index used is a common one, but has not been shown for winter and summer separately before, as far as we are aware, so no citation is possible.

*Line 94-95: authors clearly refer to the "simple connections" between global dioxide emissions, increases in global average temperature and severe impacts on people. However, it might be useful to underline more the importance of "talking through the links" when using the approach described in the paper, to avoid an over-simplification of the physical processes.*

We have added the word '*communication*' before '*tool*' on L95 to emphasise the talking aspect.

*Overall, the manuscript could provide an interesting contribution.*

**Reviewer 2:**

*The article presents an approach to communicate changes in local climate hazards via a physically plausible causal chain. The approach is illustrated for a river basin, and connects various observations starting from GHG emissions and ending with peak flow river measurements. While the single connections in the causal chain are not novel per se, I have not seen it being laid out in such a comprehensive way. It certainly is a very sensible and helpful strategy in illustrating and communicating robust observed connections of global CC to the local level, without giving much room to misinterpretations.*

*The paper fits all the characteristics of a GC Insights paper, it is written well and concise, it includes one figure.*

We thank the reviewer for their careful reading of the manuscript and constructive comments.

*I have only some very minor comments:*

*Title: While the paper discusses in some parts other (anthropogenic, non-climatic) aspects that lead to changes in climate risk (a compound of hazard, vulnerability and exposure) and justifies why the used river basin is less susceptible to such, I was wondering if it would be beneficial to change the word "risk" with "hazard" in the title.*

We agree that most of the discussion refers to the hazard component. However we consider that 'risk' is a more appropriate word for communication purposes, which is the primary aim of this study.

We have added the following sentence at the end of the paragraph starting on L26:

*Although we are using the general term 'risk' for communication purposes, we mainly focus on the hazard component; changes in exposure and vulnerability are also relevant, but are not emphasised here.*

We have also added additional text to one sentence in the paragraph starting on L80: *The complexities can include the role of antecedent conditions and evaporation, and catchment storage in groundwater or soils that could dampen extreme rainfall increases. Similarly, direct anthropogenic modifications, such as reservoirs, river engineering or floodplain development can directly influence flood hazard but may additionally influence exposure and vulnerability.*

*Line 23: "changed decisions" -> "change decisions".*

Agree – this will be changed.

*In my opinion, this work provides a valuable contribution.*